# Modeling the Land Cover Change in Chesapeake Bay Area for Precision Conservation and Green Infrastructure Planning

**Xinge Zhang [1], Kenan Li [2,\*] , Yuewen Dai [1] and Shujing Yi [1]**

1   Weitzman School of Design, University of Pennsylvania, Philadelphia, PA 19104, USA;
    zhaxinge@upenn.edu (X.Z.); yuewen@design.upenn.edu (Y.D.); shujingy@design.upenn.edu (S.Y.)
2   College for Public Health and Social Justice, Saint Louis University, St. Louis, MO 63103, USA
\*   Correspondence: kenan.li@slu.edu

**Abstract:** This study developed a precise land cover model to predict the shifts from pervious to impervious surfaces in the Chesapeake watershed. Utilizing 1 m resolution longitudinal land cover data from the Conservation Innovation Center (CIC), our model achieved impressive balanced accuracies: 98.96% for Portsmouth, 99.88% for Isle of Wight, and 95.76% for James City. Based on the analysis of feature importance, our model also assessed the influence of local socioeconomic and environmental factors, along with their spatial lags as represented by natural splines. These outcomes and findings are crucial for land use and environmental planners, providing them with tools to identify areas of urban expansion and to devise appropriate green infrastructure strategies, while also prioritizing land conservation. Additionally, our model offers insights into the socioeconomic and environmental drivers behind land cover changes. Its adaptability at the county level and reliance on widely available data make it a viable option for other municipalities within the Chesapeake basin to conduct similar analyses. As a proof-of-concept, this project underscores the potential of precision conservation in facilitating both land preservation and the advancement of green infrastructure planning, thus serving as a valuable resource for policymakers and planners in the region.

**Keywords:** land use; Chesapeake watershed; precision conservation; impervious surfaces; green infrastructure

## 1. Introduction

Remote sensing has witnessed growing interest within the realm of urban planning and spatial analysis, due to its commendable cost-effectiveness and technological reliability [1–3]. Presently, we possess multiple instances of global urban products from the Global Human Settlement Layer (GHSL) [4] and Global Urban Footprint (GUF), which can provide significant insights into understanding urban land change at regional and global scales. However, these global products, while useful for broader level analysis, fall short in detailing land cover change at the local level [5]. Due to the fact that these global frameworks aim to capture a variety of urban land change contexts, they often miss out on the drivers required for local-level accuracy. Especially when extracting land cover information through object-based image analysis (OBIA), the effectiveness and accuracy largely depend on the variability and uncertainty in the OBIA process.

In recent years, a notable trend has emerged in the field of land use development. It involves the creation of comprehensive regional databases through the skillful fusion of local data by planning analysts who possess an in-depth understanding of the local conditions. These regional databases are of substantial significance, as they provide valuable insights and data for local development planning [6–8]. However, the transformation of such extensive data on high-fidelity urban land use predictions into actionable policy guidance remains an ongoing challenge [9]. Multiple phases of the policy cycle are rarely tackled with remote sensing data, as many studies remain at the conceptual level, rarely providing recommendations for actions [10,11]. Especially at the local site level, the drivers

for implementing real change in the environment lie in the incorporation of conceptual and technological advances in policy, strategic thinking, and design [12]. For example, when planning the design for best practice management, it is essential to have precise information on permeable and impermeable surfaces down to the scale of roads to guide pond design. Multifaceted patterns of urban land use present a formidable obstacle, impeding policy-makers from deriving direct and effective insights from land use alterations. In contrast, the reclassification of land use types followed by comparative analyses appears to offer a more tenable avenue for discerning the nature of urban land use/cover changes [13].

Existing studies and examples of reclassifying land use types and conducting comparative studies to understand land cover changes have played a crucial role in advancing our knowledge of environmental transformations. Before leveraging the remote sensing dataset, research efforts have focused on reclassifying surveyed land use data and using the context-based model such as the SLEUTH model to understand the transition from agricultural to urban land use in rapidly growing cities [14]. With the increasing availability of aerial images and Lidar data, there is more research using the integration of data from multiple sensors results in an enhanced land cover product and then classifying the product for specific use [15–17]. These applications aim to understand the correlation between neighborhood characteristics and land cover, such as adult mosquito abundance data to inform critical public health concerns [18], urban heat [19,20], and tree cover and social equity [21]. These understandings have notable repercussions on public health, water resources, spanning community and site-level planning, and land use regulation [22].

The Chesapeake Bay watershed plays a crucial role as both an ecological and economic asset, supporting a range of ecosystems and sustaining vital industries like agriculture, tourism, and fisheries. Nonetheless, the US Environmental Protection Agency has listed the Chesapeake Bay as impaired due to its non-point source loads of nutrients and sediment (Chesapeake Bay Program, 2000). Knowing the probability of land conversion from agriculture, wetland, or forest (resource lands) to residential, commercial, or industrial use (built) will guide the development of practical alternatives and contingency plans related to Bay trends and indicators [23]. However, the region faces escalating environmental challenges due to the combined effects of sea-level rise and land subsidence. It ranks as the nation's second-most vulnerable area to flooding and storm surge, second only to New Orleans [24]. Central to confronting these issues and driving forward climate adaptation and mitigation planning is the accurate prediction of land cover changes, particularly the transition from pervious to impervious surfaces.

Existing studies have delved into modeling pervious surface changes to facilitate climate adaptation in urban planning [25]. These investigations recognize the pivotal role of land cover alterations, particularly the expansion of impervious surfaces, in exacerbating urban heat island effects and other climate-related challenges [26]. Researchers have employed various approaches, such as remote sensing and geographic information systems, to monitor and predict changes in land cover, emphasizing the need for increased green spaces, green infrastructure, and urban forestry to improve the urban ecology [27], avoid extreme heat exposure [28], and flooding [29]. Moreover, these studies often stress the importance of precise land cover data for accurate climate modeling and adaptation strategies, underlining the potential of innovative data fusion techniques to enhance the accuracy of land cover maps for urban areas [30,31].

There is a significant gap in the current models regarding the study of changes between impervious and pervious surfaces, as they predominantly focus on detection rather than prediction [32]. Models that utilize land cover data for prediction often center on urban growth, employing conventional methods like the SLEUTH models. However, these models lack the necessary precision in identifying changes between pervious and impervious surfaces, resulting in suboptimal accuracy [33,34]. This shortcoming is largely due to the under-utilization of reclassified land cover databases as reliable predictive tools, particularly for previous land cover changes on a cellular level.

To overcome these issues, our model leverages high-resolution land cover prediction data, significantly enhancing the accuracy and reliability of our predictions. We adopt the random forest algorithm, enriched with carefully selected socioeconomic features (Table 1) and environmental factors inspired by the SLEUTH models. This advanced approach positions our model as a notably accurate and effective tool in comparison to existing research in this field.

**Table 1.** Independent variables for model testing.

| Factors | Features |
|---|---|
| Socioeconomic factors [1] | Population change |
| | Pct of white change |
| | Unitchange |
| | MedHHIncchange |
| Environmental factors | Slope |
| | Dem |
| | Soil Type |
| Spatial Lag factors [2] | $x_n s_1$ |
| | $x_n s_2$ |
| | $y_n s_1$ |
| | $y_n s_2$ |
| Land cover factors | road |
| | canopy |
| | water |

[1] For the most precise population change data, we use block group-level census data. [2] Spatial lag factors are derived from the x and y coordinates within the raster dataset. Specifically, $x_n s_2$ and $x_n s_1$ represent non-linear transformations of the original x coordinates, and the 2,1 represents the degree of freedom of the transformation.

Historically, health warnings regarding Perfluorobutane sulfonic acid and GenX chemicals have been closely linked with the region's agricultural practices, posing threats to the well-being and lives of residents [35]. The significant role of spatial data becomes evident in highlighting the cooling effect of green buffers on runoff, even as impervious surface proportions increase due to urban development [36]. With strategic green buffer distribution, the impact of impervious surface expansion on urban growth can still be mitigated. Established urban greenways can reduce stormwater runoff and combat the urban heat island effect caused by impervious surfaces [37,38]. Parks, wetlands, and rooftop gardens can reduce the adverse impacts of impervious surface expansion on the city's microclimate, biodiversity, and flood resilience [39]. Additionally, research studies have shown that urban green buffers along urban streams can significantly reduce the input of pollutants into aquatic ecosystems [40].

## 2. Materials and Methods

### 2.1. Study Area

This study focuses on three distinct counties within the Chesapeake Bay watershed, each representing unique developmental contexts (Figure 1). Portsmouth serves as an urban archetype, characterized by its dense residential, commercial, and industrial zones. James City County exemplifies the suburban context, encompassing a mix of rural, suburban, and urban developments, featuring diverse landscapes including forests, wetlands, and historic sites. Lastly, Isle of Wight County embodies the rural aspect, predominantly marked by agriculture, forestry, and extensive natural habitats. By encompassing these diverse counties, our objective is to create a comprehensive and universally applicable model for predicting land cover changes across a variety of regional development scenarios.

In partnership with CIC, we procured longitudinal and high-resolution land use and land cover data aimed at driving "Precision Conservation" strategies in the realms of climate adaptation and mitigation planning. This initiative holds significant potential, par-

ticularly through analyses that offer applicability across partner counties and municipalities spanning the Planning District and the wider Chesapeake Basin. Our analysis focuses on three emblematic counties of the Hampton Roads Planning District Commission (HRPDC) Zone [41]—Isle of Wight, James City, and Portsmouth—each representing rural, suburban, and urban landscapes.

The choice of our study area originates from a specific necessity. In the Hampton Roads planning district, discussions of prospective developments raise the possibility of spatial overlap with the Chesapeake Bay Preservation area, which is subject to preservation acts and conservation regulations. Notably, construction projects may necessitate special permissions to safeguard existing green infrastructure, while the conservation of local wetlands and habitats remains a paramount concern. To proactively address this, early identification of potential developments is vital. Leveraging a machine learning model to predict the probability of land cover transitions to imperviousness at a fine resolution, we can accurately anticipate future development and allocate resources accordingly [42]. The outcomes of our predictive analysis in this chosen study area will offer invaluable insights that can guide targeted regulatory actions and funding allocation for green infrastructure.

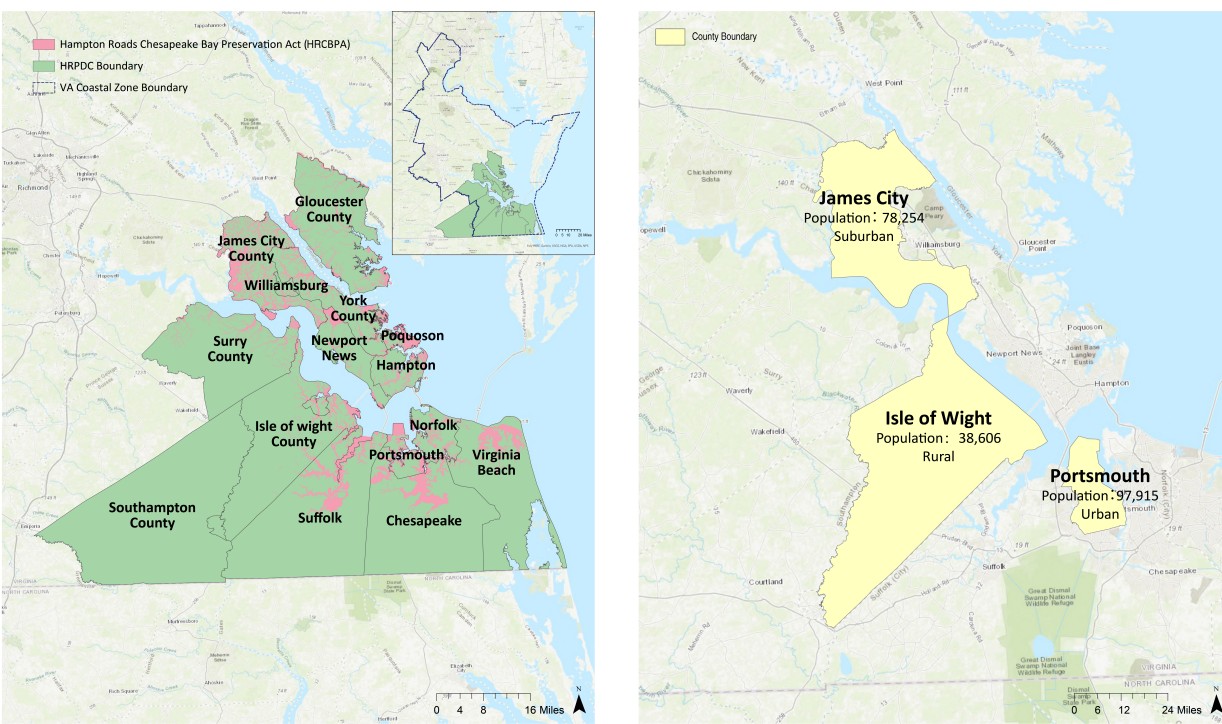

**Figure 1.** Hampton roads planning district commission (HRPDC) zone and the study area.

*2.2. Data Source*

We obtained high-resolution land cover data for 2013/14 and 2017/18 from the Chesapeake Conservancy [43]. This comprehensive raster dataset boasts a remarkable 1 m accuracy, offering 900 times more detail than the commonly used 30-meter resolution in the National Land Cover Dataset. Such a granularity is crucial to capture subtle changes in land cover. Chesapeake Conservancy, U.S. Geological Survey (USGS), and University of Vermont Spatial Analysis Lab (UVM SAL), with funding from the Chesapeake Bay Program (CBP), produce such 1 m resolution land cover and land use datasets spanning the Chesapeake Bay watershed regional area (encompassing 206 counties and an area over 250,000 km$^2$). These data offer a foundational, authoritative, and transformative perspective on the region's landscape and its holistic management.

The production of the CBP 1 m "land cover" data involves the delineation and classification of image objects derived from aerial imagery (primarily the National Agriculture Imagery Program, NAIP), topographical details derived from LiDAR, and other ancillary data. Land cover represents the surface characteristics of the land with classes such as impervious cover, tree canopy, herbaceous, and barren. In contrast, "land use" represents how humans use and manage the land with classes such as turf grass, cropland, and timber harvest. Producing land use from land cover data requires a variety of ancillary datasets combined with spatial rules that leverage the contextual information inherent in the land cover data.

The CBP land use/land cover (LULC) data are distinctly named to highlight their amalgamation of cover and use classes, such as extractive barren and solar–herbaceous. These classes are critical to understanding the impact of human activities on the Chesapeake Bay. For example, one singular land cover class, like herbaceous vegetation, can encapsulate both the highest polluting land use (e.g., corn production) and the least impactful ones (e.g., natural succession). LULC data contextualize land cover classes for decision-making, such as informing outcomes in the Chesapeake Bay Watershed Agreement and serving as the basis for developing the next generation of watershed and land change models.

Additionally, the CBP 1 m LULC data have over 50 unique classes, providing more categorical context than the 13-class CBP land cover data or the 17-class NLCD data. This detailed classification scheme is necessary to ensure that these data are widely applicable for supporting data-driven decision-making by the Chesapeake Bay Program and other regional stakeholders.

Land cover classification includes pervious surfaces, such as tree canopies and shrubs, which allow water to infiltrate the ground. In contrast, impervious surfaces encompass categories like roads and structures that prevent water infiltration, leading to increased runoff and potential flooding issues. Although water and wetlands are often considered impervious surfaces, in this study, we classify them as pervious surfaces due to their dynamic nature, interaction with groundwater, floodplain connectivity, and the critical functions of wetlands in water storage and infiltration.

The temporal resolution of the data is set at a four-year interval. This decision stems from the variability in regions covered by NAIP aerial imagery. To maintain uniformity in temporal statistics and error analysis across the entire watershed area, this level of temporal granularity represents the best achievable consistency. Nevertheless, for purposes such as detailed policy implementation, robust management control, and effective pollution monitoring, this level of detail typically proves adequate. This project is one of the efforts trying to assist the new water quality parameters (WQPs) which have traditionally been analyzed and monitored through sampling and laboratory testing, and are expensive, labor-intensive, time-consuming, and not suitable for large-scale analysis. In our study, we try to utilize the data to assist other possible use cases.

To comprehensively understand how land cover change is influenced by various environmental, social, and economic factors, we acquired additional data from the following sources: we procured a the digital elevation model (DEM) with a 1 arc-second resolution from USGS, collected soil data from the Web Soil Survey, and extracted Census tract-level data (population, white population, household unit, median household income) from the American Community Survey (ACS) for the years 2014, 2018, and 2021.

*2.3. Data Processing*

Under considerations of computational efficiency and scalability, we resampled the original dataset to 500,000 data points for model building and then tested the fitted model on the whole dataset with 850,000 data points for subsequent predictions. We employed geo cross-validation at the block group level to ensure the robustness of our model and avoid overfitting.

Performance evaluation and model selection were conducted using the confusion matrix for binary threshold setting. We calculate the accuracy, sensitivity, and specificity of the models and compare their performance based on the confusion matrix.

Figure 2 shows the whole workflow of our modeling process. We ran the land cover prediction model based on the Equation (1) composed of social economic factors, environmental factors, and original land cover types that affect the land cover change. All the factors are calculated based on the 10 m × 10 m raster cell and detailed in Table 1. All the variables are calculated for the overtime changes for every four years.

$$Y = \text{Socioeconomic factors} + \text{Environmental factors} + \text{Spatial Lag factors} + \text{Land Cover factors} \tag{1}$$

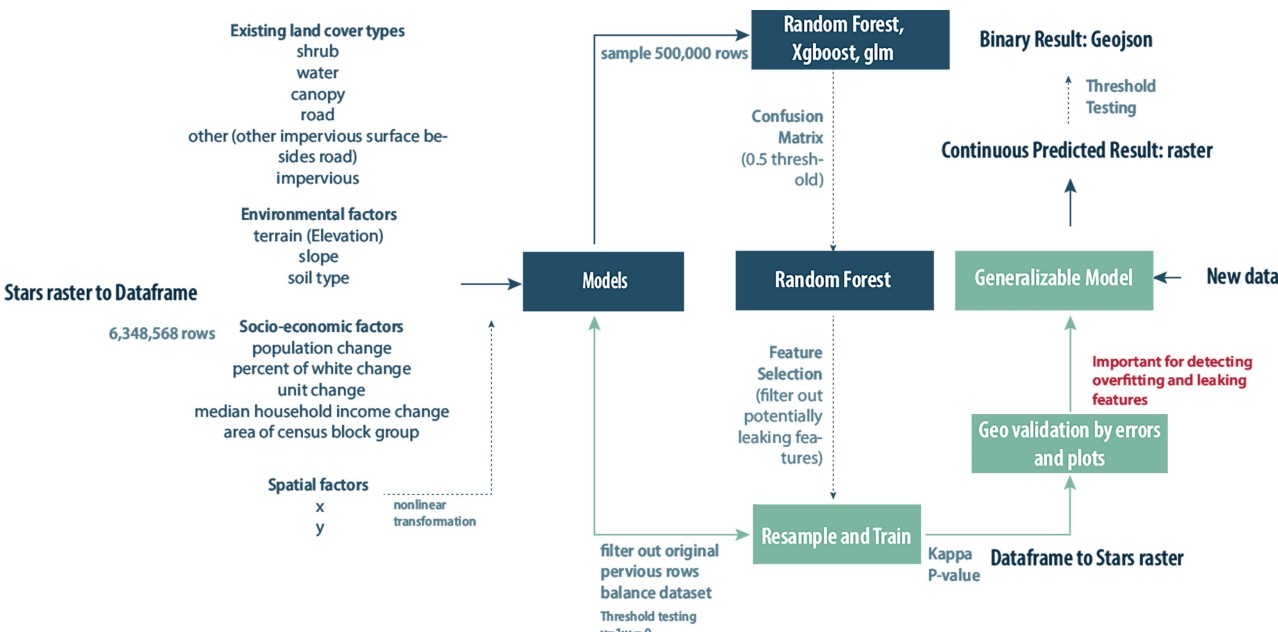

**Figure 2.** Workflow of the data processing and modeling process.

### 2.3.1. Unit of Analysis

To maintain consistency in our analysis and facilitate future investigations, we are-interpolated all remote sensing data to a 10 × 10 m resolution, which serves as our primary analytical unit. This resolution strikes a balance, providing sufficient detail for planning purposes (for example, a 10 m resolution will not affect the resolution of minor roads in the image, while significantly minimizing noise and reducing the dataset size, thus enabling a faster calculation process).

We have opted to build our model at the county level, a decision driven by two key advantages. Firstly, this approach effectively translates the broad-scale management of watershed ecological health into practical policy implementation scales. Secondly, it enhances the accuracy and efficacy of predictive outcomes [44]. In the realm of land use and conservation, a common concern revolves around the potential impact of land preservation on economic development. Conducting cell-level (spatial dimension: 10 m × 10 m) planning at the county level aids in coordinating different administrative units under the guidance of overarching conservation policies. For example, in the context of our project, the Chesapeake watershed has established policies for land use within its designated protection zones. However, the execution of these policies is the responsibility of individual administrative regions. With the assistance of our models, each administrative unit can make adaptive adjustments based on their unique land use conditions. Through our analysis, it becomes apparent that socioeconomic indicators wield significant influence over the overall predictions. This highlights that, while collaborative and indicator-based control at the watershed scale is feasible and logical for comprehensive environmental health planning

and management, personalized planning, modeling, and control for each specific region prove more effective and meaningful for precise planning, detection, and management.

### 2.3.2. Equation for Feature Engineering

Equations (2)–(5) show the calculations of the social economic factors being used in this study. Equations (6)–(9) calculate the values of natural splines at specific locations for creating smooth surfaces of the input variables.

Social economic factors:

$$\text{Population Change} = \frac{\text{Total Population of 2018} - \text{Total Population of 2014}}{\text{Census tract Area}} \tag{2}$$

$$\text{Pct of White Change} = \frac{\text{Total White Population of 2018} - \text{Total White Population of 2014}}{\text{Total White Population of 2018} \times \text{Census tract Area}} \tag{3}$$

$$\text{Household Unit Change} = \frac{\text{Total Household Unit of 2018} - \text{Total Household Unit of 2014}}{\text{Census tract Area}} \tag{4}$$

$$\text{Median Household Income} = \frac{\text{Median Household Income of 2018} - \text{Median Household Income of 2014}}{\text{Census tract Area}} \tag{5}$$

Spatial lag factors: $X, Y$ is cell coordination in the raster dataset, $x_n s_1$ is the value of the natural spline at $X$, $\alpha$ represents the intercept or constant term; $\beta_1$, $\beta_2$ are coefficients associated with the polynomial terms, $(X - a)$ represents the deviation of the predictor from the left endpoint $a$ of the segment.

$$x_n s_1 = \alpha + \beta_1(X - a) \tag{6}$$

$$x_n s_2 = \alpha + \beta_1(X - a) + \beta_2(X - a)^2 \tag{7}$$

$$y_n s_1 = \alpha + \beta_1(Y - a) \tag{8}$$

$$y_n s_2 = \alpha + \beta_1(Y - a) + \beta_2(Y - a)^2 \tag{9}$$

### 2.3.3. Spatial Pattern Analysis

According to most urban growth models [45], there are change drivers and constraints in the land use development and urban growth. Specifically for the impervious surface, the impervious cover change correlated with specific land use like the road, canopy, and water. The original land use data can be treated as cells and evaluated by the distance to the existing land cover type. As our data products are built upon existing land use cover data, and given that the land use types affecting the transition from permeable to impermeable surfaces are both dispersed and represent a relatively small proportion of the dataset, we have adopted a strategy that relies on focal calculation (as detailed in Table A1) based on individual features. This approach effectively facilitates spatial interpolation, making the most of the spatial efficacy of data points that constitute a minority. It also serves the purpose of preventing data leakage.

During our focal calculation, we achieved a controlled distance decay of the spatial impact of various land elements through multiple iterations of focal mean calculations. By incorporating a range of morphological parameters, we successfully identified subtle spatial variations within our data.

Figure 3 illustrates the outcomes of spatial feature calculations within 1000 m × 1000 m land units. It demonstrates the varying spatial influence of different features on 'y'. This scale of 1000 m × 1000 m is commonly employed in urban planning, allowing us to encompass diverse land use categories and observe fine-grained changes in land use types, such as roads and water bodies, within urban areas [46].

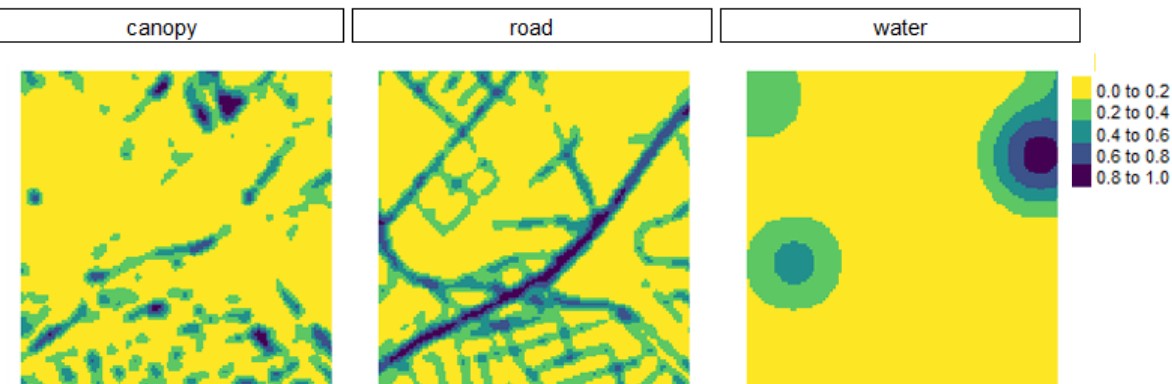

**Figure 3.** Examples of focal calculation of original land cover features. The plots presented here depict a spatial dimension of 1000 m × 1000 m with a cell size of 10 m × 10 m.

### 2.4. Modeling

We modeled the probability of land cover change from pervious to impervious as a function of socioeconomic factors, environmental factors, spatial lag factors, and proximity to existing land cover types. We refined our final model through a series of steps, including experimenting with different feature combinations, rigorous model testing, and careful parameter tuning to optimize results.

To ensure computational efficiency and scalability, we down-sampled the original dataset to 500,000 data points for model building and then fitted the selected model back to the whole dataset for future predictions. We divided the dataset into two subsets: 0.75 for the training set and 0.25 for the validation set. We employed geo cross-validation at the block group level to ensure the robustness of our model and avoid overfitting.

#### 2.4.1. Model Selection

Recognizing that different types of models perform well on different datasets, we experimented with three model types to predict land cover change. Random forest, XGBoost, and binomial generalized linear model (GLM). These models were chosen due to their ability to handle complex interactions and nonlinear relationships within the data.

After evaluating the balanced accuracy of each model (Table A2), we selected random forest as the most suitable model for our analysis, as it demonstrated the best accuracy. Random forest is an ensemble learning method that builds multiple decision trees and combines their results to improve overall accuracy and stability. This model is particularly well-suited for handling high-dimensional and noisy data, making it an ideal choice for predicting land cover changes in our study area.

#### 2.4.2. Model Re-Sampling

In model selection, we acknowledged the challenge of overfitting due to an extremely unbalanced dataset and feature leakage, impacting our model's change detection accuracy despite the high overall accuracy. We refined our approach by filtering for original land cover data and achieving balance at a 1:10 (y = 1: y = 0) ratio, with multiple training rounds for stability. (We compared our results after sampling for a balanced dataset, and we tested the ratio of y = 1/y = 0 = 1:3 or 1:10, by comparing the results between two smaller counties. We used 1:10 for our models). To minimize the impact of random results generated during the down-sampling process, we implemented a more robust approach by resampling and training the model 2-3 times for each county. We then selected the best-performing model based on the Kappa statistic and the p-value for the [Acc > NIR] comparison (Table A3). This method ensures a more reliable model selection that better represents the underlying patterns and relationships between land cover changes and various factors across the different counties.

### 2.4.3. Geo Cross-Validation

Instead of randomly splitting the data into folds as in traditional cross-validation, geo cross-validation considers the spatial distribution of the data. Block groups are a suitable choice as validation units since they represent spatially contiguous areas. We assessed the model's generalizability by calculating the mean absolute percentage error (MAPE) for each census block group. The results Figure 4 show that, in Portsmouth, our model has a low MAPE for the majority of neighborhoods, with slightly higher MAPE values observed in some block groups located in the southern region. In James City, the MAPE is relatively consistent across the block groups, with only one exhibiting a higher MAPE that could be considered an outlier. When we referred to Figure A5, we could notice that the main source of errors may come from the household units change in the area, which may be the result of new construction. These findings suggest that our models are indeed generalizable and can effectively capture the patterns of land cover changes across different block groups within the counties.

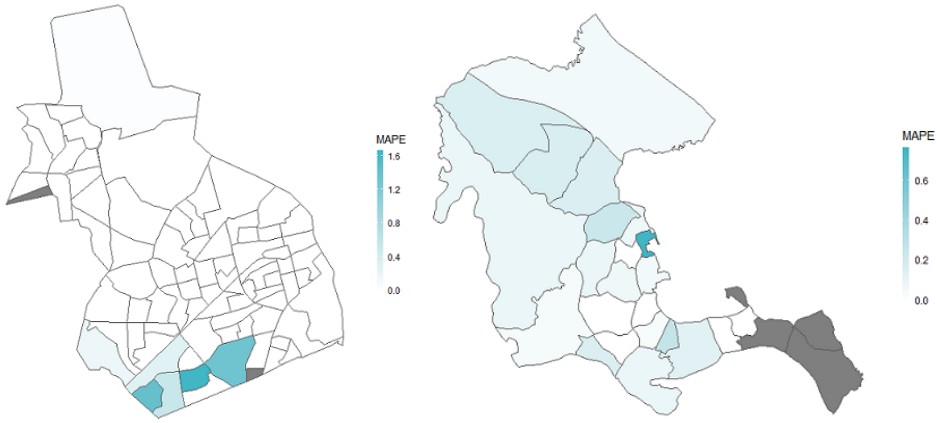

**Figure 4.** MAPE geo-validation comparison between Portsmouth and James City.

We further enhanced the model performance by optimizing the threshold to achieve the ideal balance between sensitivity and specificity. Given the inherent exiguity in land cover that may change (1), we not only offer a recommended binary prediction but also provide a continuously categorized result based on the probability distribution. This approach offers planners more meaningful insights for decision-making.

## 3. Results

In this study, we connect our findings with previous research and hypotheses [47–49], highlighting an efficient and reliable workflow for testing and modeling apervious and impervious land cover changes. The selected random forest model demonstrated a high balanced accuracy of approximately 98%, indicating its strong generalization across various counties and land cover types.

The model was trained every three to four years (based on NAIP data availability). Despite the model training process being time-intensive, the model prediction does not require too much computational time and allows for replication and scalability. The training part could also be optimized by migrating the model to the cloud platform and utilizing parallel computing. Benefiting from existing land use data of the Chesapeake watershed, our model eliminates the need for additional financial investment in infrastructure development for modeling land use changes in this region [50].

### 3.1. Counties Context Comparison and Influence Analysis

Land cover changes are closely linked to the pre-existing land cover type, as seen in Figures A1–A3, which, in turn, is significantly influenced by the developmental stages of the counties. Table 2 illustrates the economic and population conditions of different counties in our study area. Pervious surface includes canopy, shrubs, low vegetation, wetlands,

and water, impervious surface includes barren, road, structure, and others. After reclassifying land cover to 0 (pervious surface) and 1 (impervious surface), we use the 2018 value minus the 2014 value and obtain 1 as those that change from pervious to impervious surface, 0 as those that don't change, and −1 as those that change from impervious to pervious surface. Then we reclassify 1 stay as the land cover change from pervious surface to impervious surface from 2014 to 2018, 0 and −1 all as 0 for other situations.

**Table 2.** Social-economic features of the study counties

| County Names | Portsmouth | James City | Isle of Wight |
|---|---|---|---|
| Land Area (sq mi) | 46.75 | 178.72 | 362.86 |
| Impervious Area (2018) (sq mi) | 13.9 | 15 | 15.2 |
| Impervious Percentage | 29.80% | 8.40% | 4.20% |
| Population (2018) | 95,311 | 74,153 | 36,372 |
| Population Density (people/sq mi) | 2038.7 | 414.9 | 100.2 |
| Median Household Income (USD) | 48,577.89 | 88,701 | 74,591.75 |
| Percentage of White Population | 41.40% | 80% | 75.20% |
| Population Change Rate | −0.70% | 6.20% | 2.40% |

Figures A1–A3 also illustrate that, in Portsmouth, the median household income, percentage of white population change, and total household unit change exhibit a different pattern in areas where land cover changed from pervious to impervious between 2014 and 2018 (1) compared to the other situations (0). In James City, all five factors show different patterns across the two categories, but the percentage of white population change varies the most. In the Isle of Wight, the percentage of white population change, total population change, and total household unit change exhibit a different pattern in areas where land cover changed from pervious to impervious between 2014 and 2018 (1) compared to the other situations (0).

It is evident that Portsmouth represents a highly developed and densely populated urban area, James City is in a suburban phase with rapid economic growth and substantial population influx, while the Isle of Wight is characterized by gradual rural development. For James City and Isle of Wight, tree canopy is the land cover type that undergoes the most change, while for Portsmouth, low vegetation experiences the most significant change. Spatially, there are large portions of Portsmouth that are federal shipyards (non-taxable real estate for the city), plus a significant amount of commuter traffic flowing over those arteries. This also greatly affects the distribution and changes in the overall pervious and impervious areas in the county.The ability to achieve stable accuracy rates across all three counties indicates the reliability and generalizability of our modeling approach. This also adds to the importance of predicting each land cell instead of predicting the numeric percentage change in the land.

### 3.1.1. Factors Analysis

Figure 5 indicates that spatial lag factors play a crucial role in land cover change prediction for all three counties. For James City and Isle of Wight, slope is also a significant feature, whereas its importance is reduced in Portsmouth. This difference can be attributed to the urban nature of Portsmouth, where landforms influence urban growth less.

In Portsmouth, social and economic factors such as the percentage of white population change and median household income demonstrate greater significance in predicting land cover change. This suggests that urban growth in Portsmouth is more closely tied to socioeconomic factors, reflecting the distinctive characteristics of each county and the need for tailored land cover change modeling approaches.

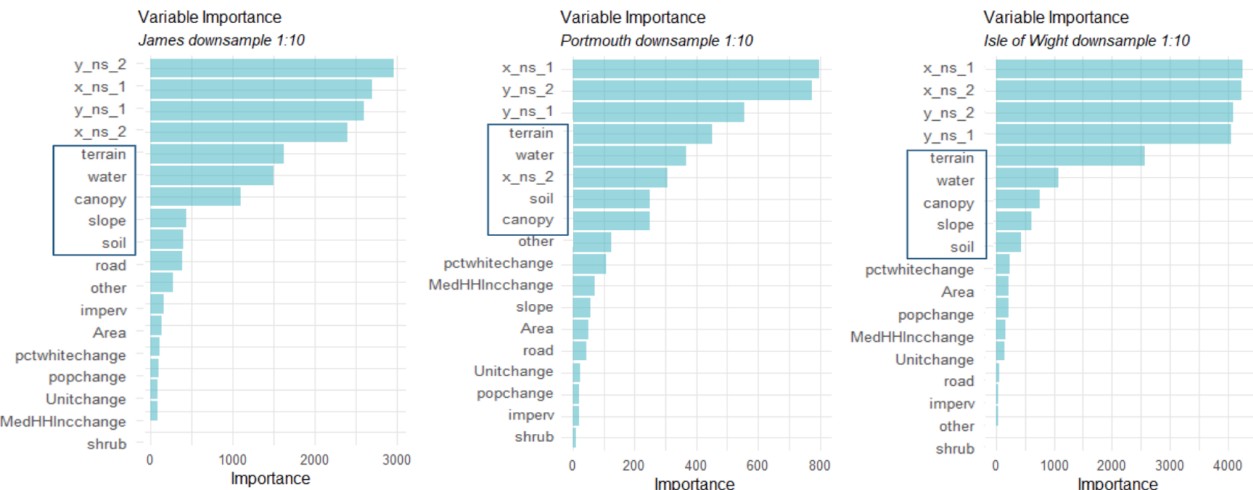

**Figure 5.** Variable importance. Feature importance is calculated based on the decrease in impurity or Gini index.

### 3.1.2. Prediction and Error Analysis

To better visualize the results, we select a sample square with a size of 1000 m × 1000 m in Portsmouth (Figure 6). This grid size allows us to magnify the scale and effectively visualize the changes in land use. By comparing the result in such a detailed scale, we can tell that errors in the prediction are likely to be around the true value, indicating the effectiveness of the model. Moreover, the model has identified new areas of land cover change. This suggests that the model successfully detects changes that were not previously observed, which can be valuable for land management and conservation efforts.

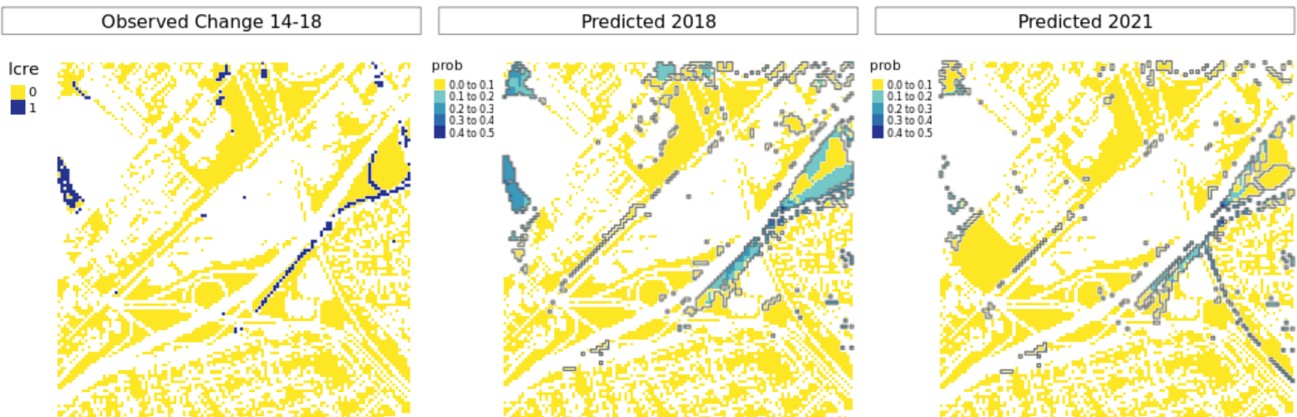

**Figure 6.** Observed changes and predicted changes of a sample square. The leftmost figure illustrates the land use changes in 2014 and 2018, while the middle figure represents the predicted results for 2018. The figure on the right displays the forecasted outcomes for 2021.

In Figure 6, based on the predicted probability, we classify the cells into two categories: changes and no changes, which are delineated using gray lines. We also filtered out areas where the land was originally impermeable marked by grey boundaries, meaning they could not transition from permeable to impermeable land use types.

To provide planners with a more intuitive and clear understanding of the predictive results and their impact on defining the boundaries of the actual conservation area, we divided the final continuous prediction values into five different risk levels based on their proportions. Figure 7 highlights the areas with a higher likelihood of transitioning from pervious to impervious land cover. In James City County, these changes are projected in regions with steeper slopes and proximity to existing development, particularly in the

northern and southeastern parts. In Portsmouth County, waterfront zones in the eastern area and midtown region are anticipated to experience this conversion. In Isle of Wight County, the pattern of such changes appears more scattered, reflecting diverse potential development across the region.

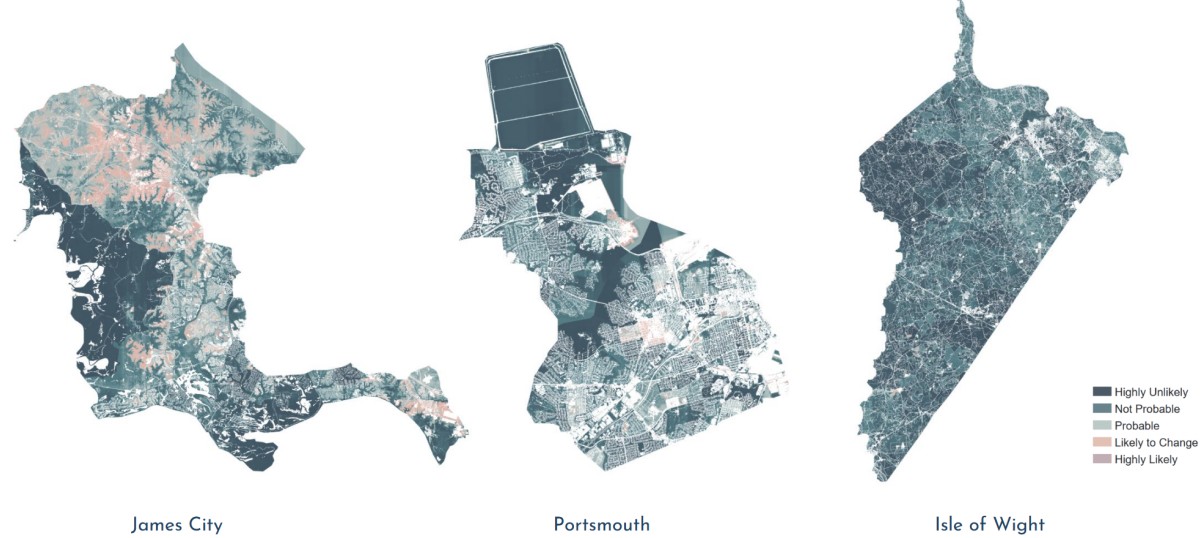

**Figure 7.** Predicted risk for land cover change.

### 3.2. App Application

For a better dissemination of our results, we developed a web application for showcasing our findings (https://tinyurl.com/precisionforecast, accessed on 28 January 2024). At the core of this application is an interactive map that allows users to view the probability of each county's region transitioning from permeable to impermeable in the future by selecting various layers. The probability interval is divided into five equal parts based on its highest value, with the high probability change areas highlighted in red. Users can further click on the "check layer" to view specific information about these high-probability conversion areas, including their past land cover types and the specific numerical values of their probability of conversion. The high-risk areas are marked in red, making them easier to locate rapidly. Users can also select other relevant information from a drop-down list, such as complete land cover maps from 2014/2018 and census tract data. This information can assist decision-makers in balancing their decisions on whether to protect certain areas or allow development based on potential impacts and the region's historical land cover changes.

## 4. Discussion

### 4.1. Novelty of the Study

Our study demonstrates the efficacy of satellite-derived data products by building a local land cover model. We compared three different types of tree-based classifiers with various parameter settings and identified the best model for the Chesapeake Bay Area. Inspired by the SLUETH model [33], we carefully selected the socioeconomic and environmental drivers that would best fit the land cover model and evaluated their importance through our random forest model. Also, by employing statistical categorization, the originally intricate land use classifications can be simplified into two categories, thereby enhancing predictive accuracy. Our model represents a significant advancement in regional environmental management and planning. Existing models (outlined in Table 3), while comprehensive, often lack the resolution [35] and specificity required to accurately reflect the unique ecological, topographical, and socioeconomic characteristics of the Chesapeake Bay area.

Table 3. Comparison of research tasks and results [32–34,51–56].

| Task | Data | Resolution | Model | Accuracy |
|---|---|---|---|---|
| LULC Classification | Landsat-8 OLI | 30 m | Random Forest | 96.01% |
| Urban Land Expansion | Landsat-8 OLI | 30 m | Cellular Automata and Markov Chain | $90.0 \pm 1\%$ |
| Land Use and Land Cover Change | Landsat (TM) 5, (ETM+) 7 and (OLI) 9 | 30 m | Neural Network | 93% |
| Impervious Surface Detection | Landsat Time Series Data and Google Earth Imagery | 5 m | Bayesian-STSRM and STCISM | 90–95% |

Additionally, this study also created a well-documented regional case study focusing on the environmental health of the Chesapeake Bay area and the similarities and differences in urban land use changes at different stages of development. The study shows that the proportion and distribution of impervious surface change varies as urban development progresses, as evidenced by the predicted results from Portsmouth, where the distribution of urban permeable and impermeable is concentrated along several major corridors as well as around the harbor, and in James City as well as the Isle of Wight. Change throughout the city is largely consistent with the suburban growth characteristics of suburban development. The spatial pattern of emerging impervious surfaces correlates with housing growth, showing a general mix of decentralization and some degree of concentration along the roadway network throughout the study area. This is indicative of a major feature of the suburbanization process throughout the metropolitan area. There is no doubt that these increases in urban impervious surfaces have dramatically altered the landscape structure and pattern. Especially in areas such as the Chesapeake Bay, where urban ecological resources, as well as ecological health, need to be protected, an understanding of this area of development will go a long way toward the implementation of overall regional ecological conservation policies and targeted pollution control.

Our research will become a strong support for urban health management. In a series of prior research and practices within this region [57–59], it has been observed that impervious surfaces play a significant role in assessing wetland health [60] and quantifying the impacts and effects of regional water temperature changes. In areas focused on advancing the total maximum daily load and utilizing best management practices to control nonpoint source pollution, this method can aid in assessing regional risks. The high-resolution predictive result will also provide crucial data support for green infrastructure planning [61], not only for maintaining the urban ecosystem but also for site-level street tree planting initiatives. The framework piloted here could also help improve the understanding of the natural–social dynamics of the landscape change especially for carbon cycle and carbon absorption [62], thereby facilitating complex approaches to environmental management and sustainable development.

The technical framework developed for the current study can be easily applied to other urban areas. Our work provides a comprehensive workflow, detailing each step from data acquisition and processing to model implementation and validation, serving as a valuable guide for similar studies in other local regions with their own social, economic, and environmental characteristics. All the tools we use are open-sourced so that any organization that replicates this analysis could do this without charge [63]. Within the current context of abundant watershed-level planning and conservation datasets, it is not difficult to imagine the greatly increased availability of high-precision land use datasets at the watershed scale in the future [64]. This greatly enhances the reproducibility of our research for using the predictive pervious and impervious land cover change to inform urban planning and policy-making, contributing to the improvement of urban environmental quality and the well-being of residents [65]. Nowadays, governments and organizations possess large-scale databases and corresponding planning policies that support comprehensive

urban planning initiatives. However, when it comes to the community level, there is a recognized importance in providing policy and planning guidance. Nevertheless, the actual implementation of these policies lags behind. To address this, we create a user-friendly Digital Earth platform that visualizes ecological and planning processes, offers suggestions, and allows for community input which would be highly beneficial for individual cities, allowing the public to openly evaluate scenarios and alternatives.

### 4.2. Uncertainties and Limitations

Due to the great imbalance of the dataset, the sensitivity of the model to invariance is significantly higher than the sensitivity to change, suggesting a user-centric model application based on specific use cases that determines the thresholds for categorization of the predicted results. Figure A4 shows an attempt at balancing the performance by varying the thresholds. Our methodology holistically tends to the extensive conservation needs of the Chesapeake Bay and simultaneously supports local administrative decision-making processes. By accurately identifying regions ripe for interventions, our study stands as a strategic guide for resource prioritization and the adoption of sustainable land management methods. The presence of errors or biases within the land cover data introduces a layer of uncertainty that significantly impacts the accuracy of our analysis. These potential inaccuracies can originate from various sources, including data collection methods, misclassifications, or inherent inaccuracies within the original data source. Evaluating the quality and potential biases in the land cover dataset becomes a formidable challenge, particularly when ground truth data for direct comparison are lacking. Moreover, our modeling approach has its own set of uncertainties. Models, as simplifications of reality, may inherently fall short in encompassing all factors that influence our study. Omissions of crucial variables or interactions within our model framework further contribute to these limitations, which, in turn, can impact the accuracy of predictions and our ability to explain observed outcomes. Additionally, a notable discrepancy in geographic units between the land cover data and socioeconomic data introduces another layer of uncertainty. Constructing a model at a spatial resolution different from the data source complicates our ability to precisely interpret the influence of varying geographic units, such as the contrast between 1m x 1m cells and 10m x 10m cells, on our analysis and its resulting outcomes. These interrelated challenges underscore the importance of cautious interpretation and transparent reporting of limitations within our research.

## 5. Conclusions

This study is based on the latest high-resolution land use data product for the Chesapeake Bay region. It involves targeted data modeling and the development of a web tool data product, addressing the scarcity of data tools in the realm of data products and practical policy formulation. This effort holds significant importance for precise land conservation and green infrastructure planning.

In an era of rapid iterations of data products, there is an abundance of predictive databases established with substantial financial resources. However, models and platforms that truly assist in translating data into effective policy decisions are often insufficient. Researchers focused on database development tend to emphasize data accuracy but may lack an understanding of the actual scenarios involving costs and data utilization. Planners, on the other hand, often lack awareness of the analysis possibilities and the potential for data products. This study successfully serves as a communication bridge between these two groups, facilitating the effective transformation of data into practical policy formulation by various government departments. However, we must also realize that our study is based on the broader context of concentrated efforts and funding in the Chesapeake Bay area to collect data and develop high-precision land-use datasets. In our research, we have found that, although most watersheds and conservation areas have their own funding and unified datasets, collaboration among conservation organizations and planning departments in formulating and implementing policies is relatively limited. Currently,

a significant challenge in large-scale remote sensing for effective land-use extraction is the need for fine-scale corrections in various regions. Only on this basis can we conduct effective and meaningful predictive analyses based on these data products. Also, governments and conservation areas are establishing relevant data departments and developing corresponding data products. However, the workflow for using data products and the modes of cooperation between organizations also require research and exploration. (See our organization workflow proposed in Figure A7).

Furthermore, the interactive web-based data visualization product not only benefits planners in policy formulation and planning analysis but also enhances the understanding of regional land use changes among the general public. It fosters environmental awareness and may even encourage public participation in future land use planning efforts. As ways for achieving this were not outlined in the reviewed literature, we set out here to outline key entry points to integrate remote sensing in urban governance giving structural advice on data handling, capacity building, and inclusion into existing legal frameworks, which needs more research and studies [66].

**Author Contributions:** Conceptualization, X.Z.; methodology, X.Z.; software, X.Z.; validation, X.Z., Y.D., and S.Y.; formal analysis, X.Z.; investigation, X.Z.; resources, S.Y.; data curation, X.Z.; writing—original draft preparation, X.Z.; writing—review and editing, K.L.; visualization, Y.D.; supervision, X.Z.; project administration, S.Y.; funding acquisition, K.L. All authors have read and agreed to the published version of the manuscript.

**Funding:** This research was partially funded by Taylor Geospatial Institute, grant number 000321.

**Data Availability Statement:** The data presented in this study are available upon request from the corresponding author. The data are not publicly available due to privacy.

**Acknowledgments:** We would like to extend our sincere gratitude to Michael Fichman and Matthew D. Harris for their invaluable contributions to the project's conceptualization and technical guidance. Our heartfelt appreciation also goes to the Chesapeake Bay Conservancy for generously providing the dataset. Special thanks to KC Filippino, Senior Water Resources Planner, and Ben McFarlane, Senior Regional Planner, from the Hampton Roads Planning District Commission for their crucial insights into the need for precise forecasting. We are deeply thankful to Dexter Locke, Research Geographer from USDA/US Forest Service, for his invaluable technical consultation. Joshua Behr's regional insights have been instrumental in shaping our project. Lastly, we express our sincere thanks to John Landis and Allison Lassiter for suggestions and project review.

**Conflicts of Interest:** The authors declare no conflicts of interest.

## Appendix A

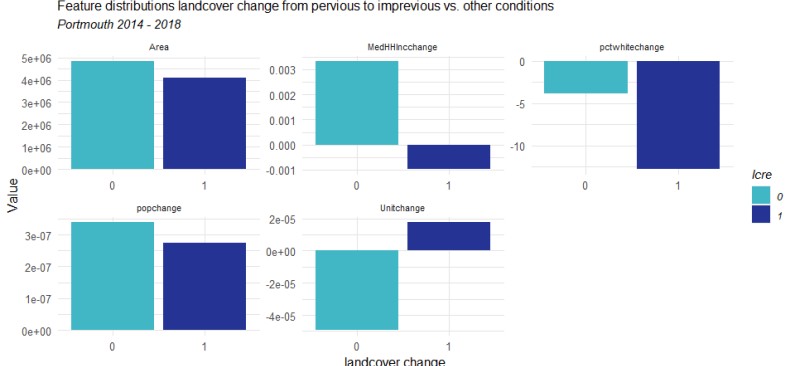

**Figure A1.** *Cont.*

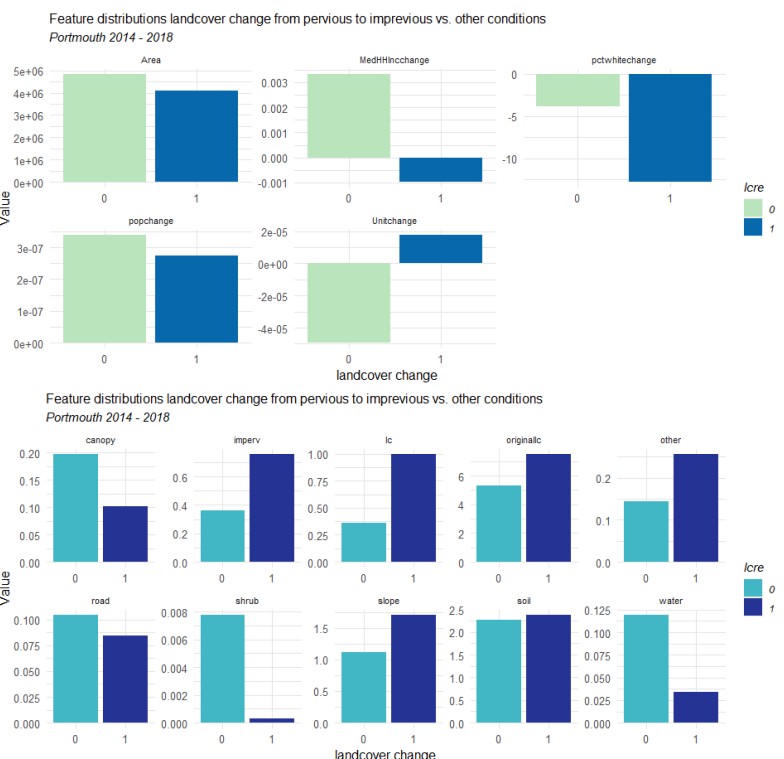

**Figure A1.** Portmouth-related feature changes across time (0 means no change or a change from impermeable to pervious).

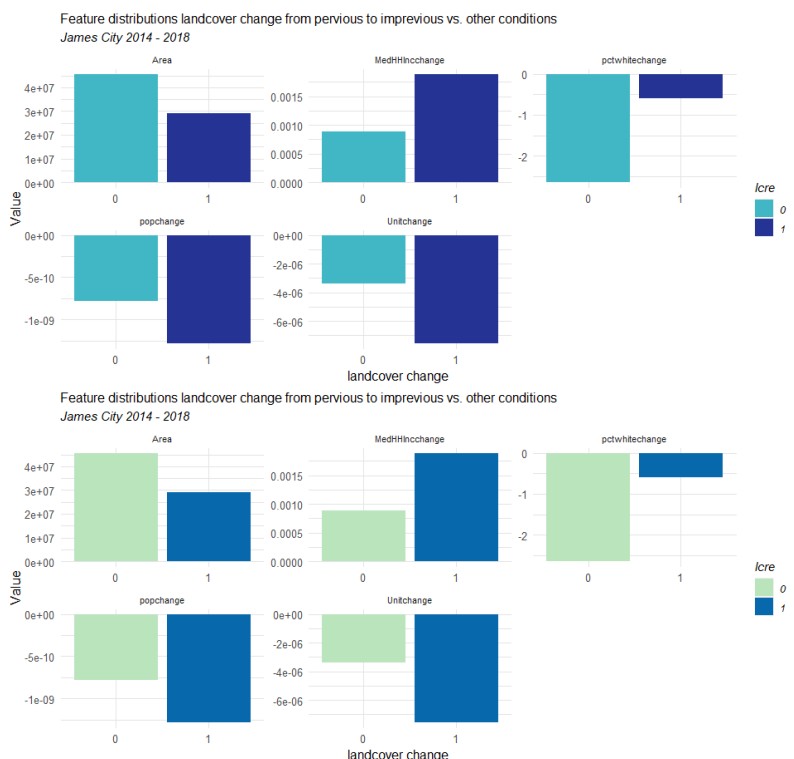

**Figure A2.** *Cont.*

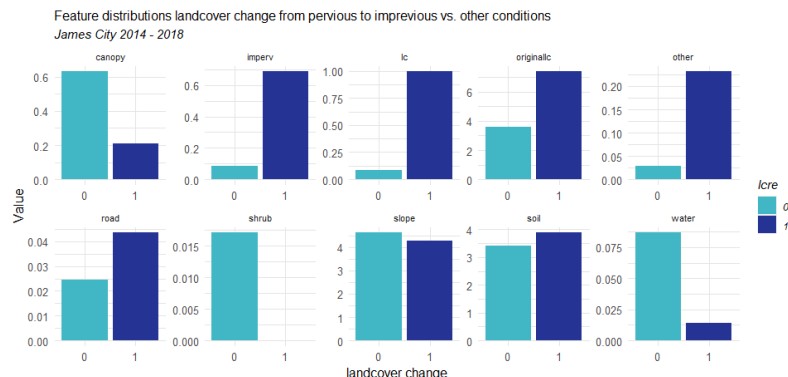

**Figure A2.** James City-related feature changes across time (0 means no change or a change from impermeable to pervious).

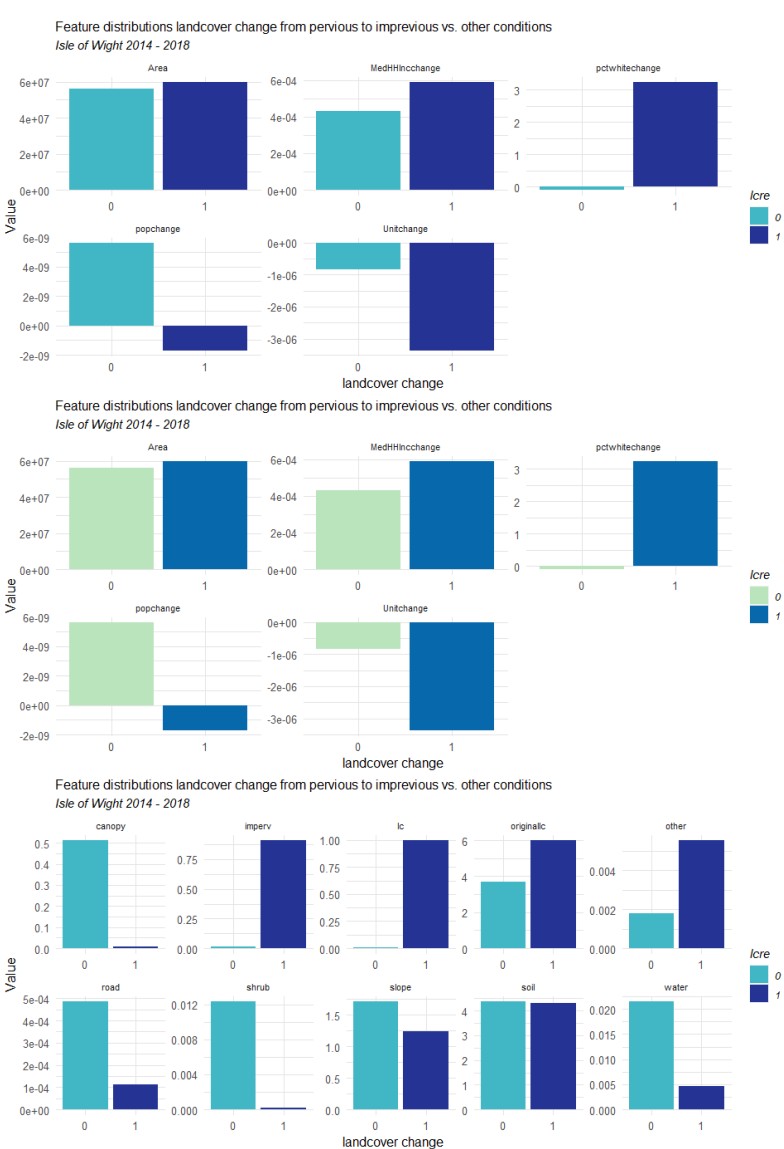

**Figure A3.** Isle of Wight-related feature changes across time (0 means no change or a change from impermeable to pervious).

**Table A1.** Focal calculation for land cover data.

| Related Landuse Type [1] | Ksize | Times [2] |
|---|---|---|
| other | 3 | 2 |
| canopy | 3 | 2 |
| road | 3 | 2 |
| shrub | 3 | 2 |
| water | 25 | 2 |

[1] Based on the experiential data obtained from interviews, that individuals utilizing this computational model should consider conducting on-site field investigations before performing calculations. [2] Focal calculation times to achieve land impact decreasing with spatial distance.

**Table A2.** Balanced accuracy for different models in various cities.

| Model | Portmouth | Isle of Wight | James City |
|---|---|---|---|
| Random Forest | 0.9896 | 0.9988 | 0.9576 |
| Xgboost | 0.9599 | 0.9928 | 0.9268 |
| Glm | 0.8905 | | |

**Table A3.** *p*-Value [Acc > NIR] and Kappa values for Portsmouth models for different rounds.

| Model | True Result as 0 | True Result as 1 | *p*-Value [Acc > NIR] | Kappa |
|---|---|---|---|---|
| Model1 result | 535,388 | 48 | 1 | 0.5464 |
| | 2777 | 1716 | | |
| Model2 result | 535,359 | 77 | $<2.2 \times 10^{-16}$ | 0.726 |
| | 1879 | 2614 | | |

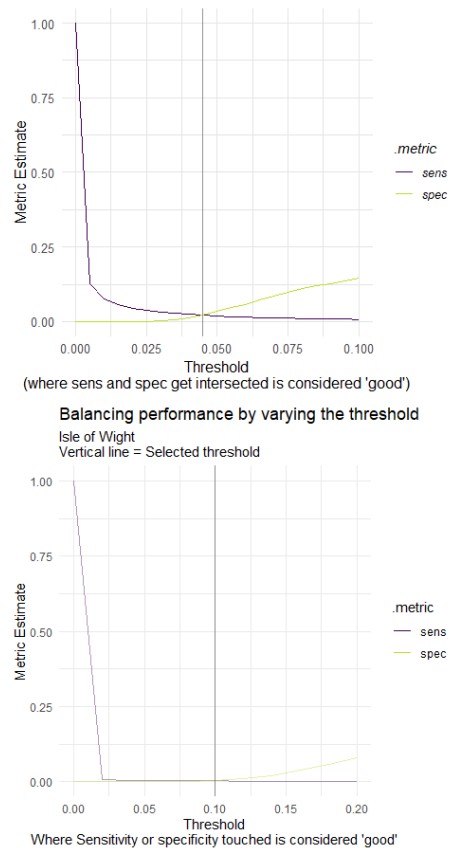

**Figure A4.** Balancing the performance by varying threshold.

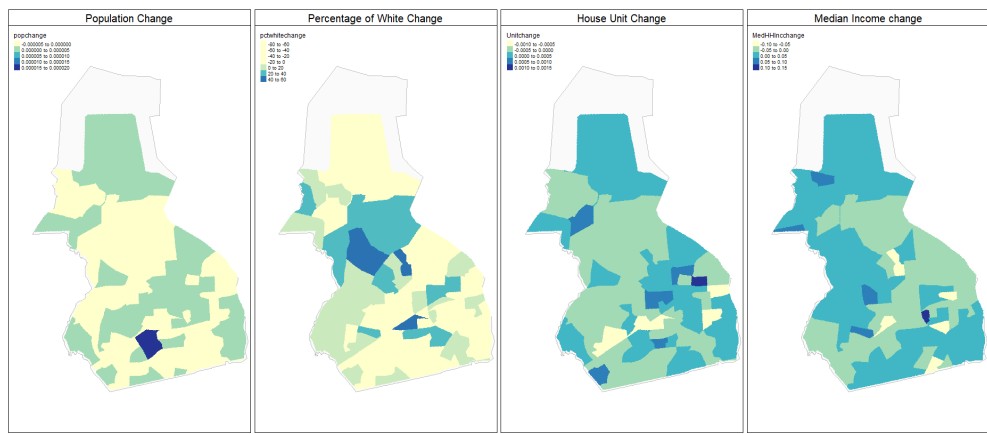

**Figure A5.** Social and economical factors in Portsmouth block groups.

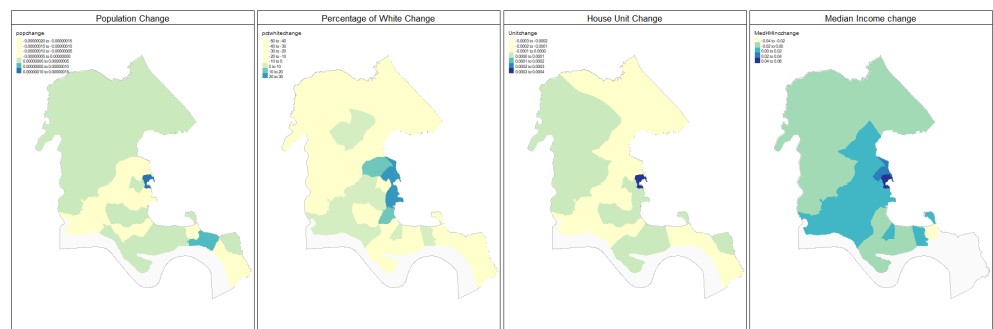

**Figure A6.** Social and economical factors in James City block groups.

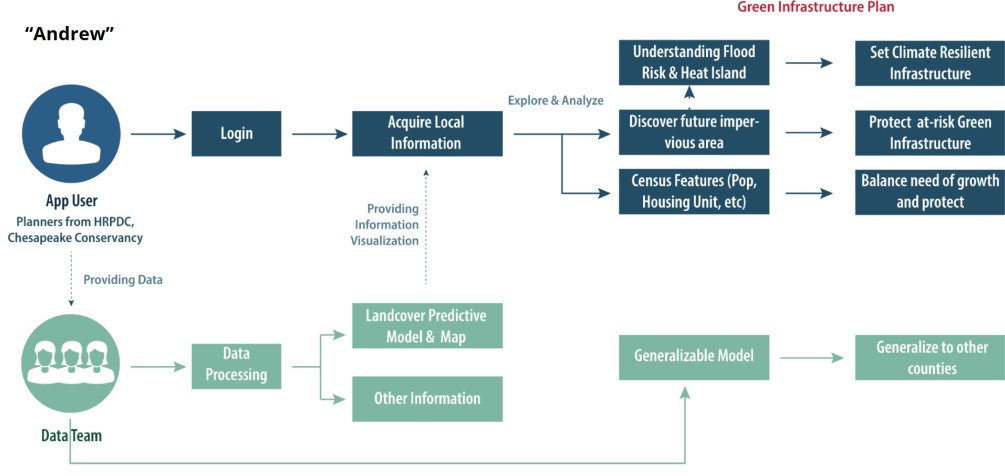

**Figure A7.** Use case proposed in our research based on the existing need and policies-making time frame. This is the proposed use case in our scenario.

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
