# Peer review of "Modeling the Land Cover Change in Chesapeake Bay Area for Precision Conservation and Green Infrastructure Planning"

_remotesensing, doi:10.3390/rs16030545_

Round 1

Reviewer 1 Report (Previous Reviewer 3)

Comments and Suggestions for Authors

As a resubmitted manuscript, upon comparing the current version with the previous one, I have observed no significant improvements. Most importantly, this article lacks innovation in the field of remote sensing, which is quite regrettable.

Author Response

Reviewer 2 Report (New Reviewer)

Comments and Suggestions for Authors

Modeling the Land Cover Change in Chesapeake Bay Area for Precision Conservation and Green Infrastructure Planning” is an interesting article. The main goal of the research was to develop a precise forecast model to predict land cover transitions within the Chesapeake Watershed.

The applied tree-based model (Random Forest) is adequate. The description of the method used and the results need improvement. I find the paper acceptable for publication with major  revision.

Some remarks & questions:

The Figure 1 map on the right has no legend; on the left map there is a buffer/an area covered by the Hampton Roads Chesapeake Bay Preservation Act, not an act itself.

The description of a study areas rows 100-108 should be moved to Section 2. Materials and Methods, 2.1. Study Area

Row 108 ‘... and urban landscapes. (Figure 1) shows our study area‘. The sentence should not start with a bracket.

Section 2.2.2. Ancillary Data is very short. I suggest including it in the preceding section and naming it "Data sources"

Row 202 (Figure 2) should not be in the backets.

Row 252 ‘With the aim of refining health standards for the overall region’ What are the health benefits? Whether it concerns the population or the state of the environment.

Section 2.3.3. Feature Engineering , every section should include same kind of description, introduction. You should not start a section with a number, equation, or figure.

Row 297 ‘The graphic below illustrates” reference to the figure is missing.

The description of Figure 3 is partly incomprehensible. Which part of the figure represents (a) present in the description.

Section 2.3.6. Model Performance Evaluation is too short; I suggest you include it in the preceding section or expand. The issues of evaluating the model and the metrics used for it are very interesting and may help in other research.

Row 323 ‘In conclusion,’ The Conclusions section is in the end of the paper.

Row 354 ‘Table 1 illustrates the economic and population conditions of different counties in our study area’. In Table one only Independent variables for Model Testing are shown, there is nothing about study area features. Maybe it is Table 2 and I suggest moving Table 2. with social-economic comparison among counties to the section with study area description.

Table 2 Impervious Area (2018) units are missing.

Rows 368-375 the information described in this paragraph cannot be read from Table 2. Table 2 presents the values of selected socioeconomic variables, including Population Change Rate, for the three areas studied, but there is no information on the possible change of permeable areas into impervious areas. There was also no information on the year the variable values were given. white population change and unit change exhibit a different pattern.

Row 369 What does "unit change" mean.

Figure 5 The caption of the figure is very extensive; unfortunately, it is not possible to find on the figure what the individual parts of the description marked with listers (a), (b), and (c) refer to. It seems that the figure caption fits better into the description of the model's operation and should be placed in the text.

Section 3.2. App application seems a bit strange. The application examines and analyses the probability of transition from permeable to impermeable areas. Since the application is publicly available and the descriptive introduction to the application contains a lot of information contained in the article, I do not understand what the authors expect. If the material has been published, what is the point of reviewing and republishing it?

Furthermore, the materials presented at https://yuewendai.github.io/myportfolio/pdf/Chesapeake_precision_forecast.html# are better, more readable, and allow to better understand the purpose and course of the study than the description in the article. It seems to me that the author(s) wanted to include too much information in one text.

The Discussion section does not compare the results achieved to other similar studies. Were the results similar, or did the model used provide better results? 

The topic is very interesting, but the presentation of the material in the article is unorganized and, in some places, abbreviated. I suggest dividing the topic into two papers, one concerning the model itself and the variables and the model performance, the other related to the impact of variables on land cover transformations.

Author Response

Reviewer 3 Report (New Reviewer)

Comments and Suggestions for Authors

Based on the manuscript, here are some suggestions for major and minor revisions focused on strengthening the methodology and addressing research gaps:

Major Revisions:

·        The introduction could be strengthened by providing more details on the research gaps and novelty of the study. Here are some suggested recent research’s:

https://doi.org/10.1080/19475683.2023.2166989                         

https://doi.org/10.1080/10106049.2023.2210532                         

https://doi.org/10.1007/s11356-023-27554-5       

·        Elaborate more on the model validation process. Details on the cross-validation methodology are currently lacking. Consider providing additional metrics beyond overall accuracy to evaluate model performance, such as AUC, precision, recall etc.

·        The class imbalance between change/no change categories can affect model performance. Consider approaches like over/under sampling or adjusted thresholding to account for this.

·        Explain in more detail how spatial autocorrelation was handled in the analysis. This is important for land cover change modeling.

·        Provide more specifics on uncertainties and limitations - sources of error, assumptions made, data gaps etc. and how these affect the methodology and results.

Minor Revisions:

Improve explanation for choice of model parameters like 1:10 ratio for change/no change. These details will strengthen justification.

Consider adding visualization(s) of change predictions to supplement discussion and highlight patterns.

Elaborate on scalability challenges with model runtimes. Are there ways to optimize this through parallelization etc?

Provide more specifics when comparing to other land cover change models. Identify limitations of existing methods that this methodology addresses.

Enhance discussion linking land cover conversions to planning outcomes and environment impacts to demonstrate applicability.

Here are some specific comments to help the authors improve their manuscript:

1.      The main question addressed is using land cover data to predict future impervious surface expansion for conservation planning. This is clearly stated in the introduction and abstract.

2.      The topic is highly relevant and addresses a key gap in providing localized, high-resolution land cover change data to inform regional planning needs. This is a novel contribution.

3.      The study uniquely leverages specialized land cover products for the Chesapeake Bay watershed to demonstrate a methodology for precise prediction of impervious surface conversions across diverse landscapes. This is a valuable addition enabling targeted conservation efforts.

4.      On methodology, the validation process requires more detail - elaboration on cross-validation, performance metrics beyond accuracy, and handling of spatial autocorrelation would strengthen this area. Discussing limitations and uncertainties more explicitly would also improve the approach.

5.      The conclusions are supported by the evidence presented. The model achieves high accuracy in predicting conversions. Relating predictions to planning and policy outcomes more clearly would further address the main question.

6.      The references seem appropriate and provide good contextual background for the study.

7.      The tables effectively profile the differences between the counties. Figures 3 and 5 visualize predictions well, more such figures could communicate results. Captions could be more descriptive. Discussing an additional figure on change patterns would enrich interpretations.

Round 2

Reviewer 1 Report (Previous Reviewer 3)

Comments and Suggestions for Authors

The authors have made extensive revisions this time, resulting in a noticeable improvement in the quality of the article.

Author Response

Thank you for acknowledging the extensive revisions made to our manuscript and for recognizing the improvement in its quality. Your constructive feedback has been invaluable in guiding these revisions, and we are grateful for the insights you provided.

If there are any remaining areas that could benefit from further refinement, or if you have additional suggestions, please feel free to share them. We are committed to ensuring that our work meets the highest standards of academic rigor and contributes meaningfully to the field.

Once again, we appreciate your guidance and support throughout the review process.

Reviewer 2 Report (New Reviewer)

Comments and Suggestions for Authors

I appreciate the authors' efforts to incorporate my suggestions into their revision.
But there are still some issues that need improvement.
The purpose of the study stated in the abstract is inconsistent with the further content of the article, whether the model should discover and predict factors influencing changes or predict the occurrence of changes, transformations from permeable to impermeable surfaces. Especially since the authors state:”…our objective is to create a comprehensive and universally applicable model for predicting land cover changes across a variety of regional development scenarios”.
Table 1 is placed on page 4, and is referenced in the text only on page 11, Section 3.1. I suggest changing the position of tables and figures so that they are placed below the text in which the authors refer to them. This problem occurs throughout the text. In the text there is no reference to Figure 3.
Section 2.3.2 is unnecessarily separated; its content is not supported by the authors' research or literature sources.
Section 3.1.1 does not explain how the importance of individual factors was determined.
Rows 462–464 are the same as 483–486.
Although the Discussion section has been expanded, it still lacks references/comparisons to other studies. For example, authors may claim that their approach is innovative, but point to similar studies where different approaches/solutions were used compared to which yours is better. References to literature are missing in the Discussion section.
The description of the figures in Annex A requires improvement: “0 represents change from no change/change from impervious to previous...” maybe to “0 means no change or a change from impermeable to previous...”
I maintain the opinion that the topic can be divided into two papers, one concerning the model itself and the variables and the model performance; the other related to the impact of variables on land cover transformations as presentation of the material in the article in some places is of poor quality.

Author Response

Reviewer 3 Report (New Reviewer)

Comments and Suggestions for Authors

Thank you for improving the manuscript.

Author Response

Thank you for your acknowledgment of the improvements made to the manuscript. Your feedback and insights have been instrumental in guiding these revisions, and we are pleased to hear that our efforts have met with your approval.

We remain committed to further refining and enhancing the quality of our work as needed. Should there be any additional suggestions or comments from your side, please feel free to share them.

We appreciate your support and valuable contributions to the refinement of our research.

This manuscript is a resubmission of an earlier submission. The following is a list of the peer review reports and author responses from that submission.

Round 1

Reviewer 1 Report

Comments and Suggestions for Authors

The manuscript is worthy, well-written and presents orginal knowledge in urban planning. I think it can be pubished.

Reviewer 2 Report

Comments and Suggestions for Authors

Dear authors,

Thank you for sharing your work.

Developing a precise and reliable forecasting model to predict land cover transitions in watersheds is crucial for policymakers and urban planners. This study strived to construct a research framework using RF and models to predict transitions form pervious to impervious areas.

However, this study lacks originality and fails to demonstrate the characteristics of a scientific paper, resembling more of an experimental or course report. Additionally, there appears to be insufficient evidence presented this study to support a comprehensive paper.

In the Introduction section, the authors fail to address the current study developing status to bring out their research aims, as well as the innovation of this study.

In the Study area section, Figure 1 has no north arrow and scale bars, which are recommended.

The threshold of 0.5 in Figure 2's Confusion matrix is insufficient.

The dimensions of the frame in Figure 3 remain consistent at 1000*1000 before and after calculation, while the land cover features exhibit insufficient clarity.

The Results section lacks the presentation of any figures or tables showcasing the RF classification results to substantiate the reported high classification accuracy. Figure 5 fails to provide sufficient, thus lacking persuasiveness. Images in Figure 6 are more like original remote sensing images, not a significant result support.

A Conclusion section is recommended.

Comments on the Quality of English Language

Minor editing of English language required.

Reviewer 3 Report

Comments and Suggestions for Authors

The study offers a comprehensive overview of research conducted in the Chesapeake Bay watershed. The introduction provides essential background information, and the methodology is adequately explained. In summary, while the paper presents a valuable study on land cover changes in the Chesapeake Bay watershed, addressing the following points would significantly enhance its scientific rigor, clarity, and applicability.

General Comments:

1. The paper highlights the impressive accuracy achieved by the Random Forest model employed. However, it is crucial to delve into the limitations and challenges encountered during implementation, particularly concerning the significant time commitments. Exploring how these challenges could affect the scalability and practical usability of the model in real-time scenarios is essential. Addressing these concerns would significantly enhance the study's practical relevance.

2. The paper addresses the expansion of impervious surfaces and its implications for urban areas. However, there is a notable absence of discussion regarding the spatial and temporal dimensions of these changes. Are there discernible trends observed across different years or seasons? Does the expansion follow any distinct patterns in various counties? Offering a more nuanced analysis of these aspects would significantly enhance the depth of the paper.

3. The paper touches upon the potential applications of the study's findings, yet it could benefit from a more thorough exploration of the policy implications. How can the precise prediction of land cover changes inform urban planning policies? Are there specific policy recommendations derived from the study? Incorporating practical implications and actionable recommendations would considerably enhance the paper's value and relevance.

4. The paper concludes by mentioning the replicability of the study in other urban areas. While this is a valuable point, the conclusion could be strengthened by summarizing the key findings and emphasizing their significance. Additionally, suggesting directions for future research based on the study's limitations or unexplored areas would provide a compelling closure to the paper.

5. The structure of the paper is incomplete, lacking a Conclusion section to summarize the main findings of this study.

Minor Comments:

1. The text labels in Figure 1 are too small and almost unreadable, especially the annotations in the left image. It is recommended that the author increase the font size for better visibility.

2. The formatting of the article is quite chaotic. Before line 130, the alignment is justified, but afterward, it switches to left alignment. It appears that the author has not paid sufficient attention to the details in formatting.

3. The text in Figure 2 is extremely blurry, making it difficult to review.

4. The color bars in Figures 3 and 5 should be oriented horizontally. Currently, they are compressed together, making it impossible to discern the details. Additionally, the color scheme for the color bars is poorly chosen, resulting in low contrast and visibility within the images.

5. Lines 485 to 494 should not be included in the manuscript.

6. In line 496, the number "11" before the second reference is clearly a typographical error. Furthermore, the formatting of the references is inconsistent. It is advisable for the author to carefully review and edit the references for consistency.

Round 2

Reviewer 2 Report

Comments and Suggestions for Authors

Dear Authors,

1. In the last review, I pointed out that Figure 1 lacked the north pointer and scale bar, you only modified Figure 1 and ignored the same problems in other figures.

2. In the last review, I proposed "The Results section lacks the presentation of any figures or tables showcasing the RF classification results to substantiate the reported high classification accuracy. Figure 5 fails to provide sufficient,  thus lacking persuasiveness. Images in Figure 6 are more like original remote sensing images,  not a significant result support.”

The authors simply cited "The Results section lacks the presentation of any figures or tables showcasing the RF" in their revised response.

This is a highly irresponsible act. 

Again, this article has few innovation in remote sensing.

Best regards

Comments on the Quality of English Language

Minor editing of English language required.

Reviewer 3 Report

Comments and Suggestions for Authors

The authors have not sufficiently addressed my initial concerns. Many of the provided responses are vague, and upon comparing the original manuscript with the revised version, I observed minimal changes. In their replies to my review comments, the authors mentioned detailed revisions that I could not find in the revised manuscript. Frequently, the responses were brief and lacked specificity. I strongly urge the authors to carefully adhere to the suggestions provided in the first review and make comprehensive modifications. Moreover, it would be beneficial if the authors could clearly specify in their response which particular lines  of the manuscript have been modified.